# Fabrication, Modification, and Characterization of Lignin-Based Electrospun Fibers Derived from Distinctive Biomass Sources

**DOI:** 10.3390/polym13142277

**Published:** 2021-07-12

**Authors:** Amina Abdel Meguid Attia, Khadiga Mohamed Abas, Ahmed Ali Ahmed Nada, Mona Abdel Hamid Shouman, Alena Opálková Šišková, Jaroslav Mosnáček

**Affiliations:** 1Laboratory of Surface Chemistry and Catalysis, National Research Center, 33 El-Bohouth St., Giza 12622, Egypt; mohamedkhadiga728@yahoo.com (K.M.A.); monashouman@yahoo.com (M.A.H.S.); 2Pretreatment and Finishing of Cellulose Based Textiles Department, National Research Center, Giza 12622, Egypt; ahmed_nada@hotmail.com; 3Centre for Advanced Materials Application, Slovak Academy of Sciences, Dubravska Cesta 9, 845 11 Bratislava, Slovakia; 4Polymer Institute, Slovak Academy of Sciences, Dubravska Cesta 9, 845 41 Bratislava, Slovakia; upolalsi@savba.sk

**Keywords:** banana bunch (BB), palm frond (PF), organosolv treatment, PET polymer, PLA-PHB-ATBC polymer, carbon fibers (CFs), conductive polymer (poly m-toluidine)

## Abstract

From the environmental point of view, there is high demand for the preparation of polymeric materials for various applications from renewable and/or waste sources. New lignin-based spun fibers were produced, characterized, and probed for use in methylene blue (MB) dye removal in this study. The lignin was extracted from palm fronds (PF) and banana bunch (BB) feedstock using catalytic organosolv treatment. Different polymer concentrations of either a plasticized blend of renewable polymers such as polylactic acid/polyhydroxybutyrate blend (PLA-PHB-ATBC) or polyethylene terephthalate (PET) as a potential waste material were used as matrices to generate lignin-based fibers by the electrospinning technique. The samples with the best fiber morphologies were further modified after iodine handling to ameliorate and expedite the thermostabilization process. To investigate the adsorption of MB dye from aqueous solution, two approaches of fiber modification were utilized. First, electrospun fibers were carbonized at 500 °C with aim of generating lignin-based carbon fibers with a smooth appearance. The second method used an in situ oxidative chemical polymerization of m-toluidine monomer to modify electrospun fibers, which were then nominated by hybrid composites. SEM, TGA, FT-IR, BET, elemental analysis, and tensile measurements were employed to evaluate the composition, morphology, and characteristics of manufactured fibers. The hybrid composite formed from an OBBL/PET fiber mat has been shown to be a promising adsorbent material with a capacity of 9 mg/g for MB dye removal.

## 1. Introduction

Carbon fibers (CFs) are carbon-like yarn fabrics with impressive mechanical performance and myriad functional properties. CFs have a tremendous strength, low density, stiffness, and remarkable electrical and thermal conductivity. Furthermore, they are fire-retardant and chemically stable [1,2]. With comparison to metals such as steel, CFs have higher unique modulus and strength. These fibers comprise at least 92 (wt %) of carbon content [3] acquired from the pyrolysis of stabilized precursor fibers above (200–400 °C) in air with the aid of an oxidation approach. Then, in an inert atmosphere, the infusible stabilized fibers are carbonized at a temperature of about 1000 °C to extract gases and other non-carbon elements [4]. The high mechanical efficiency makes CFs desirable for using in composite manufacturing in the form of woven textiles as well as straight or chopped fibers [5].

Presently, CFs are also applied to rockets, golf club shafts, medicine, and in the automobile industry [5]. The polyacrylonitrile (PAN) is currently the main precursor for processing CFs, and it accounts about half of the cost of production [4,6]. Petroleum pitch and regenerated cellulose (rayon) are alternative precursors. The key obstacle of using these precursors is the high cost of production that restricts delivery despite the developing request; additionally, the nitrile groups in PAN generate harmful by-products [7]. The environmental interest and related costs associated with this method can be resolved by using bio-based precursors in particular [8].

One of the most polluting resources is organic waste from agro-industries. Clean technology can be incorporated to play down organic wastes via recycling [8]. In particular, renewable raw materials consisting of biopolymers or biogenic polymers are interesting sources for handling CFs. These biogenic sources are accessible in plants that are comprised from cellulose, hemicellulose, and lignin. Analysis projects have been undertaken worldwide to investigate biogenic-based CFs [9].

Lignin is the largest source of biomass with an aromatic function and a carbon content material of more than 10%. In view of lignin’s low price and carbon-building chemical structure [10], it can be considered the most fascinating sustainable precursor for the development of CFs [11]. The amount of lignin varies between plant types. For lignocellulosic materials including banana bunch, palm fronds, bagasse, etc., there is a wide range of diversity wherein their chemical behavior will be different primarily depending on original source and extraction technique [12]. The exceptional popular delignification techniques for isolating lignin are the kraft and organosolv approaches [13]. The organosolv method involves the addition of aqueous organic solvents such as ethanol, formic acid, acetic acid, and ethylene glycol to the biomass under specific temperature and pressure conditions [14]. In general, this process could virtually fractionate the three biomass components in one stage to complete the lignin and biorefinery concept standards.

The electrospinning process has picked up much consideration in the view that it can be an adaptable method that proceeds the production of nano and submicron fibers from biopolymer solution [15,16]. Nevertheless, it is important to investigate the need to generate fibers based on new lignocellulosic biomass in the application of this method [17]. However, this may be conditioned by the lack of solvents that are capable of deconstructing the fibers to create arrangements for their components (cellulose, hemicellulose, and lignin) that have affordable electrospinning properties at the same time [18]. Lignin is a large-scale sustainable source of aromatic functionalities that promote its remarkable properties, but the electrospinning of its solutions is impeded by its complex nonlinear structure. In addition to the number of possible modifications owing to its chemical structure, blending lignin with other bio-based polymers has been of great interest due to its excessive accessibility, biodegradability, and strong mechanical properties [19]. Several research studies have focused on the combination of lignin into polyethylene terephthalate (PET), polylactic acid (PLA), and polyvinyl alcohol (PVA) to form bioplastics [20]. PET is one of the most common recycled, new, and reused polymers [21]. It has been widely used as an additive to manufacture low-cost ultrafine fibers for the electrospinning technique [22]. Plasticizers may be associated with another polymer by replacing polymer interactions [20]. This phenomenon strengthened the mobility and flexibility of the polymer by promoting the lowering of intermolecular forces, glass transition temperature (*T_g_*), and handling temperature of the blends. PLA is a biopolymer obtained from corn starch and sugarcane [23]. It is a highly brittle polymer with very low toughness [24]. Therefore, several studies are trying to explore new formulations i.e., evaporation, copolymerization, and blending to resolve this constrain [25]. Poly(hydroxybutyrate) (PHB) [26], oligomeric acid (OLA), and acetyl tributyl citrate (ATBC), which impart chain mobility, are popular plasticizers for PLA, resulting in highly accelerated ductile properties [27,28].

Various studies have focused on the conversion of precursor fibers to CFs through the preservation of fibrous morphology by the process of carbonization. Throughout the stabilization process, the fibers can soften and melt together when the temperature increases above the *T_g_* of the precursor fibers [11]. Iodine has been investigated as a treatment to improve the stabilization process of various materials [29]. Iodine is well known to make charge transfer complexes (CTCs) with electron-rich particles such as those with lone pairs and aromatic rings within the form of polyiodides [30]. Consequently, this promotes improvements of dehydrogenation and speeds up the thermostabilization process.

New surface modification technologies such as conducting polymer coating, gold spraying coating, metal oxide coating, etc., [31] have been developed for lignin-based fibers with adjustable physicochemical properties that have infinite applications in science and technology including energy storage, sensors, drug delivery, and adsorption [32]. By the in situ oxidative chemical polymerization of aniline monomer in the presence of fiber using a suitable oxidant, the conducting polymer coating of hydrophilic polymers can simply be achieved. Natural fibers including kenaf, jute, mango, and coconut fibers coated with polyaniline have validated properties in various fields [33]. Therefore, polyaniline is an appropriate prospect for natural fiber coating to meet the necessity of continuous disposable technology.

Industrial wastewater containing both inorganic and organic contaminants really impacts biodiversity, the ecological environment, and the ocean framework’s characteristic activities. Amongst these contaminants, one such pollutant is synthetic dye (methylene blue, MB), which is considered to be the foremost common and harmful water contaminant [34]. It is one of the greatest prevalent used substances for dyeing cotton, silk, and wool [35]. Physical adsorption has received attention due to its straightforward treatment technique and persistent efficiency as well as low cost amongst various water purification techniques [36]. Due to the excessive cost production of high quantities of sludge and difficulties in regeneration, the cost of water treatment has risen [37]. Natural fibers, carbon fibers, and material-based polymer composites may be impressive waste water adsorbents due to their low price, eco-friendliness, and highly stable viability in traditionally organic solvents [38].

The feasibility of isolating lignin from palm fronds and banana bunches utilizing organosolv treatment with acetic acid and formic acid as solvents and H_2_SO_4_ as a catalyst was investigated in this research. Throughout the electrospinning process, electrospun nanofibers were derived by combining lignin with either recycled polymer such as PET or biodegradable polymers such as a plasticized blend of PLA and PHB using various lignin loads. Iodine treatment of biopolymer-based fibers was carried out to enhance and hasten the thermostabilization stage. The fibers were further either carbonized at 500 °C to prepare carbon nanofibers or coated with poly(m-toluidine). The adsorption of methylene blue dye as a working pattern on tested materials was addressed.

## 2. Materials and Methods

### 2.1. Materials

In the maritime sub-tropical district (Nile Delta), palm fronds (PF) and banana bunch (BB) feed stocks were collected from neighborhood lands. The raw materials were initially rinsed regularly with distilled water to remove impurities and dust, followed by drying at room temperature. The dried samples were chopped and sieved to mesh sizes ranging from 30 to 50 mm.

All chemicals were of analytical grade and used as received without further purification unless otherwise stated. Formic acid (85%), acetic acid (85%), sulfuric acid (98%), 2,2,2-triflouro ethanol (TFE), dichloromethane (DCM), and 1,1,1,3,3,3-hexaflouro-2-propanol (HFIP) were purchased from Sigma-Aldrich (Weinheim, Germany). Polylactic acid blended with polyhydroxy butyrate plasticized with acetyl tributyl citrate (PLA-PHB-ATBC) pellets were obtained from Panara, s.r.o. (Nitra, Slovakia), while polyethylene terephthalate (PET) was received as red bottle scraps from Baldovská water (Baldovce, Slovakia). Ammonium peroxydisulfate ((NH_4_)_2_S_2_O_8_) (APS), hydrochloric acid (HCl), and iodine crystals were purchased from Fisher-Scientific (Waltham, MA, USA) as the reinforcement materials used in this study. Monomer (m-toluidine; 2-methyl aniline) purchased from Merck company (Burlington, MA, USA) was vacuum distilled before employing. Methylene blue dye (MB) was attained from RIEDEL-DE HAEN AG company (Berlin, Germany) with an assay 96%.

### 2.2. Delignification Process

The delignification process was carried out in a one-liter glass reaction vessel fitted with reflux condensers by heating the lignin source in 0.2% (*v*/*v*) H_2_SO_4_ as a promoter at 100 °C (heated by an electrical mantle) under atmospheric pressure for 2 h. Eventually, with 10 g of PF or BB, a mixture of formic acid/acetic acid with a constant (*v*/*v*) ratio of 70:30 was assorted to provide a 10:1 liquid-to-solid ratio. In order to precipitate the soluble lignin, the cooked liquors were first concentrated under vacuum to at least one-half of the initial volume applying a rotary evaporator. Then, the concentrated liquors were blended with five volumes of water and settled at room temperature for 1 h. The precipitated lignin was subsequently extracted by filtration and dried up to a constant mass at 90 °C. These samples were designated as organosolv palm frond lignin (OPFL) and organosolv banana bunch lignin (OBBL), respectively. Scheme 1 illustrates steps for the delignification process.

### 2.3. Chemical Analysis of Extracted Lignin

The content of ash was gravimetrically measured according to the TAPPI test method T-413 [39], while the content of dry and moisture matter was assessed using the standard official analysis methods of the Association of Official Analytical Chemists (AO-AC, 1990) [40].

Crude protein (CP) is an evaluation of the total protein based on a laboratory N2 analysis, which can be used to measure the total protein content in a matter through multiplying the nitrogen figure by 6.25. This is based on the premise that N_2_ is derived from protein containing 16% N_2_ (AO-AC 1984) [41].

The molecular formulas of both extracted lignin samples OPFL and OBBL were predicted by the percentages of C, H, N, and O resulted from elemental analysis [41]. In addition, their chemical composition is reflected in Table 1.

### 2.4. Processing of Electrospun Polymer Solutions

#### 2.4.1. OPFL/PLA Polymer Blend

PLA-PHB-ATBC pellets were dissolved in a single solvent system (TFE) at various concentrations (20, 30, 40, and 50% (*wt*/*v*)) embedded under magnetic stirring in water bath sonication for 24 h to eliminate particle agglomeration. All of the above-mentioned solutions have been subjected to the process of electrospinning. The optimum concentration was achieved at 30% (*wt*/*v*), which created entangled chains that were stretched during spinning, and strengthened PLA thin bead-free fibers were formed.

OPFL (0.67 g) was subsequently submerged in 10 mL of TFE at a concentration of 6.7% (*wt*/*v*) (lignin to TFE). Ultimately, PLA-PHB-ATBC solution was added to the OPFL dispersion and mixed together for 1 h either in a water bath sonicator or using a finger sonicator (UP 400S with 30% amplitude and 0.5 cycle) in an ice bath under constant stirring. The required total polymer concentration of 36.7% (*wt*/*v*) was achieved by this method. The sample is listed as OPFL/PLA/36.7%.

#### 2.4.2. OBBL/PET Polymer Blend

PET origin was extracted from waste plastic bottles crushed in a knife mill and dissolved in a binary solvent mixture composed of HFIP/DCM with ratio of (2:1) for two separate concentrations of PET (9 and 20% (*wt*/*v*)). These polymer solutions have been subjected to the process of electrospinning.

In two glass vials containing HFIP (6.7 mL), two sections of weighted OBBL (0.45 gm) were dissolved and processed for 10 min in a high-speed planetary mixer until lignin was fully dissolved. Subsequently, PET was introduced and homogenized for a further 10 min with two separate concentrations of (9 and 20% (*wt*/*v*)). After that, 3.3 mL of DCM was added to each glass vial solution to obtain a HFIP/DCM ratio of 2:1. Then, the vials were placed in an ice bath, and finger sonication was applied for 1 h. The resulted solutions were fully homogeneous and assigned as OBBL/PET/13.5% and OBBL/PET/24.5% with the total polymer concentrations of 13.5 and 24.5% (*wt*/*v*), respectively.

### 2.5. Electrospun Fiber Mats Preparation

There are four main components of the primary electrospinning apparatus: a high-voltage source (Spellman SL-150W, Bochum, Germany) that produces an electrical field between a positive-charged syringe needle and a grounded collector, 21-gauge (0.8 mm) stainless steel needle (B.Braun, Bratislava, Slovakia) with a flat end where the charged solution is pressured to stretch under its electrostatic forces, a syringe pump model NE-1000 (Era Pump System, Inc., Farmingdale, NY, USA) (5 mL), and a grounded pattern to deposit the fabricated fibers. The power supply is connected to the metallic needle by electrical wires, and there is a fairly short distance (15 cm) between the syringe tube and the target, as shown in Figure 1. The jet would be elongated by electrostatic repulsion as the solvent evaporates while electrospinning. This is followed by the thinning process that leads to the development of a micro-to-nano scale uniform fiber, which can be gathered in different orientations to create some specialized structures with distinct composition and mechanical properties. Within the unit, the temperature and relative humidity were 25 °C and 50%, respectively.

To reduce the impact of applied shear while spinning, all the checked mixed samples were held at least one hour prior to spinning. The fibers were collected on a stationary flat plate covered with aluminum foil. The syringe pump was working at a flow rate of 0.5 mL/h supplied to the spinneret by the polymer solution. For PET/9%, PET/20%, OBBL/PET/13.5%, and OBBL/PET/24.5%, the running voltage was 15 kV, while for PLA-PHB-ATBC at 20, 30, 40, and 50% (*wt*/*v*) and for OPFL/PLA/36.7%, the running voltages were 16.5 kV and 19.5 kV, respectively.

### 2.6. Handling Fibers with Iodine

The as-spun fibers (OBBL/PET/13.5%, OBBL/PET/24.5%, and OPFL/PLA/36.7%) were enclosed in a porcelain crucible and placed in a sealed glass jar containing iodine crystals. The iodine jar was kept in an air oven for 15 min at 100 °C. The system was eventually cooled down to room temperature before the fibers were detached [43]. The iodinated fiber mats have been converted into a dark brown color. They are denoted as I-OBBL/PET/13.5%, I-OBBL/PET/24.5%, and I-OPFL/PLA/36.7%.

### 2.7. Thermo-Stabilization and Carbonization Treatment

In a muffle furnace (*Shimadzu GAS CHROMATOGRAPH* (GC-14A) apparatus (used as an air oven)), the stabilization of non-iodinated (OPFL/PLA/36.7%) fiber mats was carried out heating from room temperature to 120, 140, and 160 °C at a heating rate 1 °C/min and isothermally maintained for 30 min for each temperature under a steady air flow during the entire phase. For non-iodinated (OBBL/PET/13.5% and OBBL/PET/24.5%) fiber mats, the same procedure was replicated and performed at 230, 250, and 270 °C. Then, the temperature was isothermally maintained for 1 h. The iodinated fibers were also supplied with the optimum temperature that was achieved to stabilize non-iodinated fiber mat samples.

In a tubular furnace, the stabilized (iodinated and non-iodinated fiber mats were mounted on a porcelain boat and carbonized. Throughout the carbonization phase, the heating rate was 3 °C/min. In order to preserve an inert environment, fibers were isothermally kept at 500 °C for 15 min underneath a nitrogen flow rate of 0.5 standard cubic feet per hour; then, they were naturally cooled down to room temperature.

### 2.8. Preparation of Hybrid Composite Materials

Via the in situ oxidative chemical polymerization of m-toluidine, dry as-spun fiber mats with a length of 30 mm and width of 10 mm were drawn up. In 25 mL of 0.4 M ammonium peroxydisulphate (APS, oxidant), 0.25 g of each (OBBL/PET/13.5%, OBBL/PET/24.5%, and OPFL/PLA/36.7%) nanofibers was dispersed and sonicated for half an hour. At room temperature, the polymerization process was initiated by drop-wise addition of m-toluidine (monomer) and HCl (dopant) to the above-mentioned solution for an additional 1 h under continuous sonication to obtain 0.1 M concentration of both. The weight ratio of fibers to monomer (0.5:1) was held constant in 50 mL of total reaction solution [44]. Then, the received composites were filtered and washed with repeatedly distilled water to remove oligomeric and non-treated monomers; then, they were dried at 80 °C in an air oven and labeled as OBBL/PET/13.5%/P-mTol, OBBL/PET/24.5%/P-mTol, and OPFL/PLA/36.7%/P-mTol.

### 2.9. Characterization

Elemental mapping observation of the generated samples at distinct stages was identified by using (Automatic Vario El Elementar, Device, Germany). The major elemental composition is C, H, N, and O contents. The morphology of the fabricated samples was monitored with scanning electron microscope (SEM) using QUANTA FEG 250 ESEM (Japan). A transmission electron microscope JEOL 1200FX (Tokyo, Japan) operated at 80 kV was used. Surface functional groups were analyzed through Fourier transform infrared using a KBr pellet approach on an FT-IR NICOLET 8700 spectrometer (Thermo Scientific, Loughborough, UK) in the spectral range of 400–4000 cm^−1^ along with four resolutions averaged over 40 scans. Thermogravimetric analysis (TGA) was performed to analyze the thermal decomposition behavior employing a Perkin Elmer 7 series Thermal Analysis (Perkin Elmer, USA). Samples were preheated in the nitrogen flow (200 mL/min) at 25 °C for 5 min and then heated up to 1000 °C at a heating rate of 5 °C/min. The specific surface area (Brunauer–Emmett–Teller (BET) method) and pore characteristics were evaluated by N2 adsorption–desorption at 77K with a surface area analyzer model (Quanta Chrome instruments, NOVA Automated GAS Sorption System Version 1.12, USA).

### 2.10. Liquid Phase Adsorption Characteristics

The purpose of this study is to investigate the feasibility of the treatment of aqueous solutions contaminated with methylene blue dye using as-spun fiber mats, CFs and hybrid composites. Consequently, by dissolving 50 mg of MB dye in 1 L of distilled water, a stock solution was prepared. Then, 10 mg of adsorbents at neutral pH (6.5) was administrated with MB dye (10 mL) without any external adjustment [45]. To achieve equilibrium, each sample was held in a rotary shaker at 180 rpm for 24 h at room temperature. Preliminary tests showed that a time of 24 h was sufficient to achieve conditions of equilibrium. The UV-visible absorption spectra of the supernatant solution were analyzed using a UV-visible spectrophotometer (Type UV-2401PC) in a 1 cm quartz cuvette to track the characteristic absorption peaks of MB at a wavelength of 664 nm.

The amount of adsorption at equilibrium (*q_e_*, (mg/g)) was calculated using the following equation:(1)$$q_{e}=\left(C_{o}-C_{e}\right)\;x\;\frac{V}{m}$$
where *C_o_* and *C_e_* (mg/L) are the initial and equilibrium liquid phase concentrations, *V* (L) is the volume of the equilibrium solution, and *m (*g*)* is the mass of the adsorbent.

The percentage dye removal from the aqueous solution was determined according to the following equation:(2)$$R\left(\%\right)=\left(\frac{C_{o}-C_{e}}{C_{o}}\right)\times 100$$
where *R* is the removal efficiency of the dye.

### 2.11. Mechanical Characteristics

Uniaxial tensile tests using a universal (LR10K; L1oyd Instruments, Fareham, UK) machine were conducted on as-spun fiber mats (OPFL/PLA/36.7%, OBBL/PET/13.5%, and OBBL/PET/24.5%) and their composites (OPFL/PLA/36.7%/P-mTol, OBBL/PET/13.5%/P-mTol, and OBBL/PET/24.5%/P-mTol). The load cell capacity used was 5 N. The tensile testing of tested samples was performed for 5 days in a humid environment. The fiber mats (30 mm length × 10 mm width × 0.2 mm thickness) were adhered to a paper mounting tab using an epoxy adhesive with a gauge length of 20 mm and tensile speed of 2 mm/min. When the sample was tightly grasped and cut away from both sides of the tab, the tensile test began. The average value of the tensile strength, Young’s modulus, and fraction strain was calculated for at least 3 samples [46].

## 3. Results and Discussion

### 3.1. Morphological Analysis

#### 3.1.1. Morphology of Spun Fibers

Electrospun fibers from PLA-PHB-ATBC polymer solutions with concentrations of 20, 30, 40, and 50% (*wt*/*v*) are shown in Figure 2a–d. With a steady increase in polymer concentration, the as-spun fibers with increasing diameter were arising. In addition, by increasing the polymer concentration from 20 to 30%, the beaded fibers (beads on a string structure) and fiber irregularity decreased significantly [47], and at a concentration of 30% (*wt*/*v*), they already exhibited a smooth and bead-free orientation with randomly distributed nanofibers. This sample has the best fiber morphology, adequately. The electrospun nanofibrous mats from PET solutions are visualized in Figure 2e,f. Both structures consist of uniform fibers with diameters of 410 nm and 2.1 μm for concentrations of 9% and 20% (*wt*/*v*), respectively. Table 2 reports the diameter for various fabricated samples using the Image J software (program is available on the World-Wide Web, LOCI, University of Wisconsin).

The organosolv lignin samples OPFL and OBBL manifested spherical aggregates, less defined particles with rough surfaces, and the particle diameters of 200–300 µm and 7.9 ± 4.5 μm, consequently (see Figure 3a,d, respectively).

Figure 3b,c,e, and f exemplify the impact of sonication in merging a polymer matrix within lignin. As seen in Figure 3b, the water bath sonication was not strong enough to break sufficiently the large lignin particles in the presence of the dissolved PLA-based matrix. Due to the deficiency of functional groups in polyolefines, the interfacial adhesion between OPFL and the PLA-based matrix (36.7% (*wt*/*v*)) was probably not sufficient to achieve total miscibility employing water bath sonication, so they could enter only into weak dispersion interactions [48]. However, with the use of the ice bath finger sonication, as shown in Figure 3c, a higher surface area of smaller aggregates could enable improving the interactions between lignin and matrix and interfacial adhesion within the heterogeneous blends. Thus, smoother fibers with a non-significant shift in the diameter were fabricated. This reflects the homogeneity of lignin particles inside the polymer matrix mostly during the fiber spinning process [49,50]. When a significant amount of lignin is added to the polymer blend, the mechanical properties of the manufactured fibers are ultimately influenced by the homogenization and compatibility [48].

As evident from Figure 3e,f, uniform fibers were produced by blending PET polymer with OBBL (13.5 and 24.5% (*wt*/*v*)) by applying an ice bath finger sonication. This could be attributed to good solubility of OBBL in PET solution enabling also the possible formation of hydrogen bonding between the OH of lignin and end groups of PET (ester and ethylene groups), as well as π–π interaction between the aromatic groups of lignin and aromatic groups of PET [51].

#### 3.1.2. Morphology of Fabricated Fibers after Stabilization Process

The fiber morphology of non-iodinated electrospun fibers after the stabilization process at different stabilized temperatures of various fiber mats in the air atmosphere is recorded in Figure 4 and Figure 5.

Non-iodinated OPFL/PLA/36.7%-based electrospun fibers subjected to three particular heat treatments (120, 140, and 160 °C) are shown in Figure 4a–c. The average spun fiber diameters, yielding dark brown fibers, were kept constant between 120 and 140 °C. The samples started to melt as the temperature increased to 160 °C, and fibers were mutually stacked. This is in good agreement with two melting peaks at 149 °C and 169 °C from the melting of PLA and PHB crystals, respectively, which was described for the PLA-PHB-ATBC blend in our previous work [26].

On the contrary, non-iodinated OBBL/PET/(13.5 and 24.5% (*wt*/*v*)) fiber mats showed thermal stability at 230 and 250 °C, Figure 5b,c, while at 270 °C, the fibers were fused and stacked to the crucible in black color.

Usually, oxidative thermal stabilization has induced chemical changes in the samples, such as oxidation, cyclization, cross-linking, and dehydration [52]. The presence of aliphatic and aromatic chains greatly affects the thermal treatment. In contrast with aromatic rings that are comparatively more resilient to thermal treatment, aliphatic chains have many points that can be broken, and their integrity is typically maintained throughout the pyrolysis process [53]. This could be improved by the addition of a high percentage of PET. As a result, further studies of the thermostabilization for both non-iodinated and iodinated OPFL/PLA/36.7% fiber mats were carried out by the progressive increase of temperature in the air by a rate of 1 °C/min up to 140 °C and retained at this temperature for 30 min. The thermostabilization of non-iodinated and iodinated OBBL/PET/(13.5 and 24.5%) fiber mats was carried out by the same progressive increase of temperature up to 250 °C and retained for 1 h.

#### 3.1.3. Morphology of Fibers after Their Modification

In order to prepare materials for potential application in organic dyes remediation, two types of modifications, carbonization and covering by poly(m-toluidine), were investigated (see Scheme 2).

Preserving the fiber morphology during the carbonization process is the key problem of transforming electrospun fibers into CFs. This purpose can be accomplished by introducing an oxidizing agent such as iodine crystals prior to the thermal stabilization process. Figure 6 compares SEM images of the three carbonized samples of non-iodinated and iodinated mats (I-OPFL/PLA/36.7%, I-OBBL/PET/13.5%, and I-OBBL/PET/24.5%). As seen from Figure 6a–c, all the samples prepared from non-iodinated mats were moderately shapeless masses after the carbonization process. The fibers on the collapsing lean were fused and broken as the fiber mats were heated above their melting temperature. In contrast, the iodinated fiber mats retained their fibrous structure without merging of the fibers, as revealed in Figure 6d–f. This is a consequence of the iodine, as an oxidizing agent, affecting electron withdrawal from electron-rich molecules, especially aromatic rings, to form polyiodides [43]. Through the heating process, the iodine and adsorbent can form the charge transfer complexes (CTCs) [30], subsequently producing HI and leaving behind a free radical. Thanks to the free radicals, this dehydrogenation approach promotes cross-linking reactions to improve and speed up the thermostabilization process and generates more orderly fiber morphology [43].

Some extent of melting reactions in the produced CFs were obvious in the case of the I-OPFL/PLA/36.7% mat, as illustrated in Figure 6d. They also had a larger diameter relative to CFs produced from I-OBBL/PET/(13.5 and 24.5%) mats. This can be explained by the poor interaction between iodine and the non-aromatic polylactide structure [43], which encourages less stable fiber morphology and fiber overlapping, leading to an increase in the fiber diameter, as evident from Table 2. The improvement in the color of the iodinated fiber mats through the carbonization process is illustrated in Scheme 2. After heat treatment, the color changed from mild brown to dark brown due to the highly aromatic PET structure and iodine absorbed by the lignin.

Lignin-based fibers were also used for the preparation of hybrid composites by the polymerization of m-toluidine in the presence of the fibers. Figure 7 shows the SEM images of OPFL/PLA/36.7%/P-mTol, OBBL/PET/13.5%/P-mTol, and OBBL/PET/24.5%/P-mTol fiber mat hybrid composites. As can be seen, P-mTol tended to bind to each other as clumps of granular particles with diameters of about 300 nm on the entangled fibers without any significant changes in the fiber diameter in all fabricated samples.

### 3.2. Physicochemical Characteristics

The structure of lignin is primarily reliant on wood’s nature and handling conditions, but the chemical composition and properties depend on the plant source, extraction process, and post-treatment [54].

OBBL consists of 53.7% carbon and about 36% oxygen, as shown in Table 3. For comparison, OPFL is 46.3% carbon and 46% oxygen in its molecule. The carbon content has a major impact on the efficiency of the production of carbon after heat treatment. The fundamental prerequisite for the raw material to be used as a precursor of carbon materials via carbonization is that the carbon content should be higher than 40% [55]. The elemental analysis confirmed that according to this conceptualization, OBBL and OPFL exhibited sufficiently high carbon content to be employed in the carbonization process. OBBL showed higher levels of nitrogen derived mainly from complexes of protein–lignin produced during fractionation than those of OPFL. This suggests the presence of certain contaminants during the de-polymerization process [56].

Carbon fibers derived from extracted organosolv lignin demonstrate amelioration in the carbon content after the carbonization process at a temperature of 500 °C for the samples I-OPFL/PLA/36.7% with carbon content of 54.6% and I-OBBL/PET/13.5% with carbon content of 61.7% (Table 3). This is related to the elimination of volatiles, hydrogen, and oxygen of lignin/polymer blends throughout carbonization treatment [57]. The CFs prepared from I-OBBL/PET/24.5% comprise lower carbon content (52%) and higher oxygen amount (42.2%), which is probably a result of the higher elimination of C-C bonds compared to oxygen functionalities, as was suggested by Kaur et al. during the preparation of porous carbon from PET [58].

### 3.3. FT-IR Spectroscopy

The FT-IR technique is used to provide a brief clarification on the main functional groups presented within the examined samples. Figure 8a unveils the OPFL and OBBL spectra with their produced CFs. The broad band around 3400 cm^−1^ can be ascribed to the stretching vibration of the O-H group with chemisorbed water for OPFL and OBBL. The widening of this peak is due to the –OH involved in hydrogen bonding [59]. Owing to the presence of methyl and methylene groups, the appearance of a peak at 2925 cm^−1^ shows symmetric and asymmetric C-H stretching [60]. The existence of the C=C band of alkene and aromatic-bonded oxygen groups is validated by the bands found at 1529 and 1445 cm^−1^ [61]. In the aromatic rings (syringyl units), the vicinity of about 1130 cm^−1^ corresponds to C-H bonds [62]. For the PLA spectrum, the most peculiar bands are centered at 1750 cm^−1^, which correspond to the C=O stretching vibration (ester groups) and the asymmetric and symmetric –CH_3_- deformation being aligned with the peaks at 1450 and 1360 cm^−1^, respectively. C-O-C (ether groups) bending is manifested at 1180 cm^−1^, whereas –OH bending is at 1040 cm^−1^ [63,64]. The most distinctive FT-IR bands of PET are within the range of 710 cm^−1^ (C-[CH_2_]_n_-C, skeletal stretching vibration), 935 cm^−1^ (O-C-OH bending vibration), 1290 cm^−1^ (C-O bending vibration), 1374 cm^−1^ (CH_3_ symmetrical deformation), 1415 cm^−1^ (CH_2_, bending vibration), and 1712 cm^−1^ (C=O stretching carbonyl group) in the PET polymer [65].

By comparing the spectra of produced CFs, the intensity of O-H group and chemisorbed water allocated at 3430 cm^−1^ is significantly reduced. This can be related to the hydrogen bond formation between polymers (PLA or PET) and lignin [66] in addition to the dehydration [67] and dehydroxylation [68] mechanisms as well as the volatility of water contents raised during the carbonization process [69]. The characteristic carbonyl peak intensity at 1755 cm^−1^ decreased for the CFs, I-OPFL/PLA/36.7% denoting that the PLA and OPFL are entirely miscible. This could return to the C=O and OH interactions of both PLA and OPFL. These findings agree with our study of SEM and TGA. For CFs, I-OBBL/PET/(13.5% and 24.5%), the changes in the absorption peaks in terms of the wave number and height shifting can be presumed to be due to the hydrothermal degradation of the ester group presented upon PET forming a H-bond with lignin, which intended to form an O-H hydroxyl group near 3400 cm^−1^ [70]. Basically, once PET is incorporated into as-spun lignin fibers and exposed to carbonization treatment, the chain scission and fragmentation reactions can occur. This can be accompanied by co-polymerization reactions at the lignin/PET interface.

Figure 8b visualizes FT-IR spectra of poly(m-toluidine) hybrid composites. At 1740 cm^−1^, conjugated carbonyl groups are assigned, and at 1210 cm^−1^, CO stretching (phenolic hydroxyl groups) is assigned. The peak around 1520–1480 cm^−1^ is associated to C=C stretching in the benzenoid ring. Furthermore, sharp absorption peaks are defined from 1600 to 1580 cm^−1^ (quinoid (Q) C=C), 1513 cm^−1^ (benzenoid (B) C=C) [71], and 1250 cm^−1^ (C-N vibration of aromatic amines) [72]. The peak intensity obviously decreases in both hybrid composites of OPFL/PLA/36.7%/P-mTol and OBBL/PET/24.5%/P-mTol. This reduction may be due to the collaboration between the matrix of lignin-based fibers and the chains of P-mTol. In addition, when coating lignin-based fibers, there may be a mild alteration in the chemical environment of poly(m-toluidine).

### 3.4. Thermogravimetric Analysis (TGA)

In terms of the temperature of thermal degradation, TGA curves indicate the percentage weight loss of materials. The record of thermal deterioration indicates weight loss, as well as the first derivative (DTG). Table 4 presents the critical parameters derived from these curves.

The thermal decomposition profiles of both lignin (OPFL and OBBL), as shown in Figure 9a,b, are often divided into three degradation phases [73]. The primary stage has (4.8%) weight loss that occurred between 40 and 130 °C for both lignin samples. The discharge of physically bound water to lignin is reflected in this process [12]. A more pronounced weight loss is observed with 58.2% and 55.5% perceived from 130 to 580 °C, which were respectively allocated at the second stage. The mass loss in this range is due to the thermal decomposition of the aliphatic groups and carbohydrate segments in both lignin samples leading to the formation of lateral unsaturated chains, CO, CO_2_, and CH_4_. During this temperature district, degradation of the complex structure of lignin includes the fragmentation of inter-unit linkage between phenolic hydroxyl, carbonyl groups, and benzyl hydroxyl releasing monomeric phenols into the vapor phase, moreover a few residual hemicellulose contents that can be related to the structure of lignin [74]. The syringyl units are developed primarily by ether bonds within the lignin macromolecule. At temperature below 310 °C, the ether bonds between syringyl units are easier to be broken due to their low thermal stability [75]. The final stage is between 580 and 1000 °C. Degraded volatile products from lignin comprising of alcohols, aldehyde, and acids besides the formation of gaseous products were separated from this level. DTG_max_ of the obtained samples is between 332 and 352 °C for OBBL and OPFL, respectively. Pyrolytic degradation comprising the fragmentation of inter-unit linkage releasing monomers and phenol derivatives into the vapor phase is envisaged in this domain. The observation of thermal stability mapping verified that OPFL has higher thermal stability with char yield (29%) than OBBL (25.9%). This is also reliable on the biomass source, nature, and moisture content [12].

Figure 9 gathers the TGA profiles with their first derivatives (DTG). Corresponding to the generated CFs, I-OPFL/PLA/36.7%, the profile is degraded in two stages, the first of which is ideal for ATBC plasticizer additive and PHB with maximum degradation at 289.5 °C and weight loss of 45.8% [76]. However, the second step designated between 315 and 500 °C is assigned to the volatilization of PLA and degradation of lignin aromatic rings [21]. Nevertheless, it is evident from these curves that CFs, I-OBBL/PET/13.5% and CFs, I-OBBL/PET/24.5% exhibit a one-stage decomposition behavior with DTG_max_ values of 384.4 and 430.2 °C and char yield formation of 12.6 and 20.2%, respectively. This can illustrate the hydrogen bonding interaction between lignin (OH) and PET (ester and ethylene groups), as well as π-interaction between aromatic groups of lignin and PET [77]. These interactions are more common than polyolefin combination. It is commendable to note that the increase in PET% matrix content with lignin significantly improves the thermal stability of the generated fiber mats.

### 3.5. Surface Area and Dye Adsorption Analyses

Assessment of specific surface area (S_BET_, m^2^/g), total pore volume (V_t_, cc/g), and average pore diameter (D_p_, nm) was obtained for the manufactured materials using the Brunauer–Emmett–Teller (BET) equation, and the data are provided in Table 5 and Table 6.

The specific surface area of OPFL and OBBL was determined to be 258 and 146 m^2^/g, respectively, and the mesoporous structure character could be expected. The topological defects of lignin that are affected by the biomass source may be correlated with this difference in surface properties [78]. For comparison, the electrospun fiber mats OBBL/PET/(13.5% and 24.5%) were 340 m^2^/g and 246 m^2^/g, respectively, indicating surface area 1.5 and 2.5 times larger than that for OBBL. In addition, the total pore volume has also increased. The reinforcing effect of PET on the OBBL assembly can be manifested by the increase in the surface area and the structure disorder of nanofibers, which could be introduced through the spinning process. Furthermore, a theory discussed the feature dimension of interfiber voids, and the limit of contact between fibers in the electrospun polymer network is related to the specific surface area of the network. These features have had a major effect on the cross-sectional morphologies of the fibers [79]. As a result, OBBL/PET/13.5% nanofibers had a greater surface area than OBBL/PET/24.5% microfibers.

It is noted that compared to OBBL/PET spun fiber mats, the as-spun OPFL/PLA/36.7% fiber mat encompassed a slightly smaller surface area (70.9 m^2^/g) and pore volume. This could be due to the connection of the fibers during the electrospinning process, as well as the obstruction of gas adsorption and desorption by organic salt crystals embedded on the rough fiber surface [80]. Furthermore, PLA is hydrophobic in nature, and this limits its dispersion into the polymer matrix and hydrophilic solvents, so PLA is not soluble in water. By comparison, lignin is hydrophilic due to the presence of hydroxyl groups on its surface, and its strong H-bonding makes it easily soluble in water. The inter-facial compatibility between the non-polar matrix and the greatly polar lignin surface is low, a weak filler–matrix interaction is formed, and the surface area decreased.

The most interesting feature is that the BET surface area and pore volume are found to seriously decrease after coating fiber mats with m-toluidine polymer; OPFL/PLA/36.7%/P-mTol (61 m^2^/g, 0.06 cc/g), OBBL/PET/13.5%/P-mTol (112.5 m^2^/g, 0.01 cc/g), and OBBL/PET/24.5%/P-mTol (42 m^2^/g, 0.1 cc/g). This can be caused by the occlusion of pores by grown P-mTol on the fibers surface and the restriction of dispersed gas adsorption/desorption associated to pore reduction [81].

After carbonization, both samples of CFs, I-OPFL/PLA/36.7% and CFs, I-OBBL/PET/24.5% pronounce larger surface areas of 147 and 385.6 m^2^/g, respectively. In contrast, CFs, I-OBBL/PET/13.5% showed a decrease in surface area (286.3 m^2^/g) compared to the fibers before carbonization. To clarify, the absence of graphitic crystallinity is observed during the process of carbonization [82]. As a consequence of forming the basic mesoporous structure, the resulting char becomes rich in carbon, and non-carbon atoms are removed. Additionally, the cracked surface morphology is created by the process of chemical thermostabilization. The decrease in surface area for CFs, I-OBBL/PET/13.5% can be attributed to the enormous quantities of volatile matters released during the carbonization process at 500 °C. The carbonization residues (i.e., tar) can block the pores due to incomplete carbonization. Tar, which was derived from fiber degradation, can also hinder the formation and growth of new pores, resulting in low specific surface area and total pore volume [83]. Moreover, the fibers have higher weight loss during carbonization, exhibiting a larger diameter of surface pores [84]. Consequently, the surface area decreases with smoothing the surface of the fibers.

The applicability of the prepared lignin-based fiber mats for the removal of dye from water was investigated using MB dye solution. As shown in Table 5, the adsorption of a massive molecule such as MB dye was independent of the surface area determined by BET. Except for the OPFL/PLA/36.7% fibers with the lowest lignin content, both OBBL/PET fibers were able to adsorb the MB dye. Lignin fibers contain hydroxyl and aromatic rings with its framework, whereas MB dye includes nitrogen atoms within its structure, and the lone pair of nitrogen atoms produces intermolecular H-bonding with these functional groups. Consequently, some of these physical forces including intermolecular H-bonding, electrostatic interaction, as well as π–π stacking between the aromatic ring of MB and aromatic rings of both lignin and PET can work synergistically and motivate the successful removal of dye from aqueous solutions [85].

Interestingly, the carbonized samples did not lead to an improved adsorption of MB dye, suggesting that losing some of the functional groups during the carbonization process has a negative effect either on interactions with MB dye or wettability and thus the interface between the CFs and aqueous solution. The highest adsorption efficiency toward MB dye was obtained after the polymerization of m-toluidine on the fiber surface. Thus, the adsorption efficiency of 2.8, 13.1, and 17.7% was obtained for hybrid composites OPFL/PLA/36.7%/P-mTol, OBBL/PET/13.5%/P-mTol, and OBBL/PET/24.5%/P-mTol, respectively. The order is in good agreement with FT-IR spectra in which the highest signals from P-mTol were observed for the sample OBBL/PET/24.5%/P-mTol. The P-mTol can further increase the adsorption of MB dye due to π–π stacking interactions between aromatic groups of P-mTol and MB as well as possible hydrogen bonding between the amine groups in the P-mTol and MB dye structures [85,86]. Cationic MB dye adsorption onto the basic sites of poly(m-toluidine) can also occur through electrostatic interactions [87,88]. The MB dye adsorption capacity of manufactured samples was compared with those of other adsorbents stated in the literature, as revealed in Table 6.

### 3.6. Mechanical Characteristics

For the applicability of the prepared materials, their mechanical properties are important. Therefore, the tensile properties of fabricated lignin-based fiber mats and their composites were also investigated and shown in Figure 10 and Figure 11. The results are also summarized in Table 7. As described in our previous works, the brittleness of PLA/PHB blends can be significantly suppressed by the addition of ATBC as a plasticizer [27,28]. However, the introduction of lignin in the sample OPFL/PLA/36.7% provided a dramatic decrease in tensile strength (0.22 MPa), stress at break (0.2 MPa), and strain at break (15%) indicating weak plastic properties, while the Young’s modulus was 3.2 MPa. The reason could be the inherent interactions based on strong H-bonding between carboxylate groups of PLA and PHB as well as hydroxyl groups of lignin causing a decrease in the mechanical characteristics of the plasticized PLA/PHB blend.

In contrast, as can be seen in Figure 10, the tensile strength, stress at break, and strain at break of neat as-spun OBBL/PET/(13.5% and 24.5%) fiber mats were enhanced significantly higher compared to OPFL/PLA/36.5% fiber mats. This can be due to the lower extent of the hydrogen bonding between hydroxyl groups of lignin and ester groups of PET due to the presence of an additional π–π interaction between lignin and PET aromatic groups. These interactions can also minimize lignin aliphatic hydroxyl groups, helping to prevent lignin phase coalescence during polyester (PET) matrix blending [97]. According to the possible strong fiber–fiber interface packaging, the introduction of PET helps to increase the efficiency of blends, improving mechanical properties [97]. The Young’s modulus of neat as-spun OBBL/PET/13.5% fibers (64 MPa) was greater than that of OBBL/PET/24.5% fibers (23.3 MP). This is often correlated with the integration of lignin into the PET matrix, which improves the compatibility and adhesion between lignin and PET, resulting in a higher Young’s modulus and leading to more flexible characteristics of the generated (OBBL/PET/24.5%) fiber mat [98], as can be seen from Figure 11. The excellent characteristics of polyesters such as high strength, good stretchability, and durability can be taken into account [99].

The dispersion of one constituent into the other is important when producing composites, as this can have a significant impact on the determination of the mechanical properties of the composite material. A large improvement in the mechanical properties is possible if a uniform dispersion of the reinforcement filler in the matrix is achieved as well as great interfacial adhesion. This is not the case when the composite is prepared by coating of the surface of fibers by polymer aggregates. The tensile properties of the hybrid composites (OBBL/PET/(13.5% and 24.5%)/P-mTol) showed a slight decrease in tensile strength (1.5 and 2 MPa), respectively. This result points out that the polyester fabric could be weakened during in situ oxidative chemical polymerization, and that poly(m-toluidine) could penetrate the fiber, resulting in low composite crystallinity exhibiting a decrease in tensile strength [100]. Nevertheless, the elongation at break was almost the same, providing the material with handling that is sufficient for organic dyes removal applications. However, with OPFL/PLA/36.7%, a hydrogen bond rivalry arises between a protonated nitrogen belonging to the amine group within poly(m-toluidine) via hydroxyl and a carbonyl group of PLA with the addition of P-mTol. The bond between PLA and PHB was broken down according to this reaction. So, PLA brittleness was monitored throughout acidic medium polymerization [101].

### 3.7. Summary

Lignin was successfully extracted from palm fronds and banana bunches as biomass sources using the organosolv fractionation technique. The combination of lignin with either polyethylene terephthalate (PET) or plasticized polylactide/poly(hydroxybutyrate) blend enabled the production of fibers with smooth morphology by electrospining as confirmed by SEM. Pretreatment of the fibers with iodine improved the quality of the fibers after the thermostabilizing process. The best blend compositions were selected for additional post-processing either by carbonization or coating by in situ oxidative chemical polymerization of m-toluidine. The carbonization treatment at 500 °C still ensured retaining of the fiber morphology, but the achieved carbon content did not exceed 62 wt %; thus, it did not provide fully carbonized carbon fibers, as was confirmed by elemental analysis and TGA. In situ oxidative chemical polymerization was used to produce hybrid composites by coating electrospun fibers with electrically conductive polymer (poly(m-toluidine)). Several techniques including Fourier transform infrared (FT-IR), scanning electron microscope (SEM), thermogravimetrical analysis (TGA), elemental analysis, BET measurements, and tensile properties were performed to examine the structural and morphological features of the fabricated samples. The highest adsorption capacity of about 9 mg/g was reported for hybrid composite of OBBL/PET/24.5% according to the investigation of the adsorption process of MB dye for all samples tested. Furthermore, tensile testing of the OBBL/PET/24.5% fiber mat revealed that it has good mechanical properties with tensile strength and elongation at break of about 3.1 MPa and 69%, respectively. Even though more post-modification optimization experiments are needed, the work clearly demonstrates the potential of combining biomass waste as a lignin source with sustainable waste (PET) to produce fibers as a core material that can be used for environmental remediation.

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
