# Peer review of "Fabrication, Modification, and Characterization of Lignin-Based Electrospun Fibers Derived from Distinctive Biomass Sources"

_polymers, 2021, doi:10.3390/polym13142277_

Round 1

Reviewer 1 Report

In the manuscript “Fabrication and Post-treatment of Lignin-based Electrospun Fibers for Organic Dye Removal”, the Authors prepared and investigated new lignin-derived spun fibers for their potential application in remediation of methylene blue, which is the most common and harmful organic dye in industrial wastewater.

Although the subject is interesting (especially due to potential environment protection) and methods chosen for the study are appropriate, some concerns need to be addressed before the manuscript is ready for publication.

Major comments:

  1. Line 109: It should be “charge transfer complexes (CTCs)” (as in line 419) instead of “charge exchange complexes (CTCs)”. The term “charge transfer complexes” is used by Tanabe et al. (Ref. 32).
  2. Lines 147-148: The fragment “2. Materials and Methods.” should be removed. An appropriate section title is given below (line 149).
  3. Line 159: No company name is given. It should probably be “were purchased from Sigma-Aldrich (Weinheim, Germany)” instead of “were purchased from (Weiheim, Germany)”.
  4. Line 183: There is no number of the section “Chemical Analysis of Extracted Lignin”. It should probably be 2.3. Therefore, the subsequent sections should be given new numbers (lines: 196, 197, 210, 223, 247, 254, 268, 280, 295, and 313).
  5. Line 184: It should be “TAPPI test” instead of “TAPPT test”. TAPPI is an abbreviation of Technical Association of the Pulp and Paper Industry.
  6. Line 239 (Figure 1): I suggest to use “Scheme 2” instead of “Figure 1” (Scheme 2 in the present version of the manuscript should then be changed into Scheme 3).
  7. Lines 281-282: The sentence “Elemental mapping observation of the generated samples at distinct stages was identified by using (automatic Vario Microanalyzer Device).” should be corrected.
  8. Line 292: BET acronym should be explained when first mentioned (please compare line 567).
  9. Line 298: There is “MB dye (BDH)” in section 2.9., while in section 2.1. Materials (lines 166-167) one can read “Methylene blue dye (MB) was attained from RIEDEL-DE HAEN AG company (Berlin, Germany)”. This issue should be addressed.
  10. Lines 324-325: There is no section 3.1.
  11. Figure 5: The caption of Figure 5 is not clear enough.
  12. Lines 474, 524, 564, 640, and 694 (Results and Discussion): The same section number 3.3. is given to different sections: FT-IR Spectroscopy, Thermo-gravimetric Analysis (TGA), Surface Area and Dye Adsorption Analyses, Mechanical characteristics, and
  13. Line 523 (Figure 8 caption): It should be “13) OBBL/PET/13.5%/P-mTol” instead of “13) OBBL/PET/12.5%/P-mTol”.
  14. Figure 9 caption: It should be “with their produced CFs; 1) OBBL,” instead of “with their produced CFs;.OBBL,”.
  15. Figure 9 caption: It should be: “5) CFs, I-OBBL/PET/24.5%.” instead of “5) OBBL/PET/24.5%.”.
  16. Lines 588-589: I have doubts about the statement regarding the solubility of lignin in water - “lignin is hydrophilic due to the presence of hydroxyl groups on its surface and its strong H-bonding makes it easily soluble in water.”.  This issue should be addressed (for more details please see, eg, Lora and Glasser, Recent Industrial Applications of Lignin: A Sustainable Alternative to Nonrenewable Materials, Journal of Polymers and the Environment, 2002, 10).
  17. Figure 10: It should be “OPFL/PLA/36.5%” for the black curve in the figure instead of “OPFL/PLA”.
  18. Page 25: Table 6 is incorrectly positioned. It should be inserted below the caption of Figure 11 (the title of Table 6 is given in line 692).
  19. Lines 706-707: I suggest to check the sentence “Adsorption properties of all post-modified fiber mats were analyzed by BET and adsorption of methylene blue dye.”

Minor comments:

  1. Line 91: It should be “terephthalate” instead of “terephethalate”.
  2. Line 101: It should be “(PHB) [24]” instead of “(PHB)[24]”.
  3. Line 159: It should be “Weinheim” instead of “Weiheim”.
  4. Lines 162-163 and 271: The name of APS should be given in the same way - ammonium peroxydisulphate (line 162) or ammonium peroxydisulfate (271).
  5. Line 185: It should be “method T-413” instead of “method.T-413”.
  6. Line 332: It should be “nano fibers” instead of “Nano fibers”.
  7. Table 2 (left column): Bold font should be not used for PLA-PHB-ATBC.
  8. Line 397: It should be “Figure 4. Photo” instead of “Figure 4 .Photo”.
  9. Lines 552, 556, and 557: It should be “CFs, I-OPFL/PLA/36.7%” instead of “CFs,I-OPFL/PLA/36.7%”
  10. Line 614: It should be “As shown in Table 5, adsorption” instead of “As shown in Table 5, Adsorption”.
  11. Line 635: It should be “[90,91]” instead of “[91] [90]”.
  12. Text editing is suggested.

Reviewer 2 Report

This manuscript demonstrates a fabrication and post-treatment of lignin-based electrospun fibers for the adsorption of organic dyes. Although the authors made great efforts to characterize the morphology of prepared fibers and test their performance for MB adsorption, the manuscript needs a major revision because of following reasons.

  1. The title should be revised to a more specilized one.
  2. Abstract missed the quantitative presentation of adsorption performance for MB. Moreover, it should be rewritten because it only elucidated what this paper was done without little results.
  3. There are many typos and mistakes in this manuscript.
  4. The introduction is too long and tedious to be followed.
  5. The reference should be concise and more relevant papers elucidating the adsorption performance and mechanism are encouraged to be discussed. For instance, 
  6. Scale bar is missing in the SEM images.
  7. Figure 9 should be rearranged.
  8. A comparison of MB adsorption capacity with other adsorbents is missing.
  9. Table 6 and Figure 11 are overlapped.
